# Strategies for the Development of NK Cell-Based Therapies for Cancer Treatment

**DOI:** 10.3390/cells14231858

**Published:** 2025-11-25

**Authors:** Tatiana Budagova, Anna Efremova, Margarita Maiak, Dmitry Goldshtein

**Affiliations:** Federal State Budgetary Scientific Institution Research Centre for Medical Genetics, Moskvorechye Str. 1, Moscow 115522, Russia

**Keywords:** CAR-NK, hnCD16, IL-15/IL-15Rα, immune checkpoints, safety switches, clinical trials

## Abstract

**Highlights:**

**What are the main findings?**
For the generation of CAR-NK cells, the most promising approach is to use the CD56^bright^CD16^dim^CD57^neg^ NK cell population.The design strategy for CARs is based on either creating a highly specific CAR-NK (e.g., anti-CD19 CAR-NK), or a broadly acting CAR-NK with the sequence of a conventional activating NK receptor (e.g., NKG2D CAR-NK).To enhance the cytotoxicity of NK cells, it is possible to modify them with high-affinity non-cleavable CD16 and intracellular activation domains.To overcome tumor immune suppression, it is feasible to knock out immune checkpoints in NK cells.To maintain long-term proliferative activity, NK cells can be engineered to express IL15 and IL15Rα.To ensure the safety of CAR-NK cells for humans, they can be modified by safety switch.Clinical trials have confirmed the efficacy of CAR-NK cells and the genetic modifications listed above.

**What is the implication of the main finding?**
The strategy for creating a next-generation CAR-NK cell product should adhere to the following principles simultaneously: increased antitumor cytotoxicity, increased proliferative activity, and continued safety for patients.

**Abstract:**

CAR-T cell therapy is a promising method of cancer treatment, but it has some disadvantages. These disadvantages have led scientists to explore the use of safer CAR-NK cells and new genetic modifications in order to improve the effectiveness of CAR cells. In this paper, we analyze existing approaches to modifying CAR-NK cells and discuss the results of clinical trials involving CAR-NK therapies. Conventionally, approaches to NK cell modification can be divided into three main groups: strategies to enhance antitumor cytotoxicity, strategies to improve the survival of CAR-NK cells and prolong their persistence in the body, and strategies to increase the safety of CAR-NK cells. The effects of CAR-NK cells on different tumor types are presented, and the number of clinical trials involving CAR-NK cells has been increasing every year, with positive results so far. As of September 2025, all the trials are in the early 1–2 stages of research, and it is expected that the first CAR-NK product will be approved in the near future.

## 1. Introduction

Adoptive cell therapy is a promising and rapidly developing approach for cancer treatment, developed at the end of the 20th century and based on the use of immune system cells to stimulate an antitumor immune response. The first and most significant clinical success of cellular immunotherapy was associated with the use of chimeric antigen receptor (CAR) T cells for the treatment of hematological malignancies, which has led to the approval of 7 CAR-T cell products by the Food and Drug Administration (FDA) [1,2,3].

CAR-T cell therapy is a novel approach that offers clear advantages over traditional methods of cancer treatment. The use of CAR-T cell therapies has shown high efficacy against certain types of hematological cancers, as they are highly specific to the antigens present on cancer cells and can persist in the body for an extended period of time. These therapies effectively destroy cancer cells, prevent metastasis, and do not require surgical intervention. Additionally, CAR-T cells can target tumors not only directly but also indirectly by stimulating the patient’s immune system [4,5,6]. However, despite the many advantages of CAR-T cell therapy, there are a number of significant drawbacks that limit its widespread use. These include the high probability of cytokine release syndrome, cytopenia, and neurotoxicity, which can lead to serious side effects such as infectious diseases. Additionally, CAR-T therapy has several serious limitations, including its low efficacy against solid tumors due to their immunosuppressive microenvironment, as well as the high risk of tumor recurrence due to tumor heterogeneity [5,6,7]. The development of CAR-NK cell therapy has become a partial solution to the side effects associated with CAR-T cell therapy, due to its improved effectiveness against solid tumors and absence of cytokine release syndrome or neurotoxicity. Other inherent advantages of CAR-NK cell therapy include its greater safety compared to CAR-T cell therapy, as well as its innate ability to destroy tumor cells in a non-specific manner, and a better cytotoxic effect on solid tumors [8,9,10]. Recent advances in the development of genetically engineered NK cells with chimeric antigen receptors have increased their safety and anti-tumor efficacy [11]. An example is the FT522 CAR-NK cell product. The product is designed to have a *CD38* gene knockout and the expression of genes coding anti-CD19 CAR, as well as a non-cleavable high-affinity CD16 receptor, IL-15 linked to IL-15R (i.e., IL15/IL15RF, or IL15/IL15R fusion) and a synthetic alloimmune defense receptor (ADR) [12]. To date, there are no CAR-NK cell products registered by the FDA or the EMA (European Medicines Agency). However, clinical trials for such products are actively underway at various stages, exploring different approaches to the development of CAR-NK cells.

## 2. NK Cell Biology and Cancer Progression

Natural killer cells (NK cells) are effector lymphocytes of innate immunity. Their functional feature is to have direct and indirect cytotoxic effects against both foreign pathogenic targets and pathologically altered host cells [13,14]. 90% of all mature NK cells are found circulating in the peripheral blood. Compared to other types of lymphocytes, the number of NK cells in the blood stream ranges from 5% to 15%, according to various sources [13,14].

The precursors of natural killer cells are hematopoietic stem cells in the bone marrow. These cells, under the influence of various factors such as transcription factors and cytokines, differentiate into NK cells [13,15]. There are three main types of NK cells, based on the expression of different surface markers: CD56^bright^CD16^dim^CD57^neg^; CD56^dim^CD16^bright^CD57^neg^; CD56^dim^CD16^bright^CD57^pos^. The CD56^bright^CD16^dim^CD57^neg^ populations precede the other two and have low cytotoxic activity. However, they can contribute to the death of and decrease the proliferation of target cells by secreting IFNγ (Interferon γ), TNF-α (Tumor Necrosis Factor α), GM-CSF (Granulocyte-Macrophage Colony-Stimulating Factor), granzyme K and other biological molecules. By secreting various cytokines and chemokines, NK cells of the CD56^bright^ population regulate the functional activity of other immune cells. CAR-NK cells can be obtained from any cell population. When primary NK cells are isolated from peripheral blood, they are often not separated into different subtypes, and the entire volume of NK cells obtained is used. However, a younger population of CD56^bright^, CD16^dim^, and CD57^neg^ cells is considered to be the most promising. This is due to the long process of CAR-NK cell expansion, which can take two to three weeks, and the short lifespan of NK cells. We wrote about this information in the text of the article [16]. The CD56^dim^CD16^bright^CD57^neg^ population is the most cytotoxic towards target cells (infected or transformed), compared to the other two populations. This is achieved by secreting granzymes A and B, as well as perforin. This population is characterized by high expression of the ADCC (antibody-dependent cytotoxicity) CD16 receptor, low expression of the NK cell surface protein CD56, and lack of expression of the CD57 receptor. Most of these cells circulate in the peripheral blood, spleen, and bone marrow [13,14,15,17]. The CD56^dim^CD16^dim^CD57^pos^ population consists of NK cells with decreased CD56 receptor expression, decreased ADCC CD16 receptor expression, and increased CD57 “maturity receptor” expression. These cells can be described as memory cells, as they effectively respond to pathogens that they have previously encountered [14,15]. The phenotypic differences between NK cells depend not only on their stage of development, but also on their localization in the body and the immunosuppressive effects of tumor cells [18]. These effects can reprogram NK cells, making them less sensitive to pathogens and less cytotoxic. 

The antitumor response of NK cells does not depend on antigen-specific priming. Instead, it is determined by a combination of activating and inhibitory receptor reactions. As a result, NK cells are able to recognize a wide range of pathogenic cells, regardless of their origin. After activation, an NK cell forms a close connection with the pathogenic target cell, known as an immunological synapse. During this process, the NK cell secretes cytotoxic granules into the synapse, which helps to destroy the target cell and protect normal nearby cells. At the same time, the NK cell also secretes various cytokines and chemokines, which attract other immune cells to the area and activate them [13,14,15,17].

During the evolutionary process, tumor cells gain the ability to undergo uncontrolled mutagenesis, division, and growth, which helps them evade immune surveillance. This is done by disrupting the presentation of neoantigens on the surface of tumor cells, disrupting signaling pathways in immune cells, inhibiting the functions of antigen-presenting cells, synthesizing immunosuppressive molecules, and creating a tumor microenvironment (TME) that is unfavorable for immune cells [19,20]. Additionally, the tumor attracts immunosuppressive cells such as T regulatory lymphocytes (Tregs), M2 macrophages, myeloid-derived suppressor cells (MDSCs), etc. [19,20]. The disruption of the presentation of neoantigens by transformed cells is achieved through genetic modification or impaired expression of MHC1 and through impaired expression of specific tumor antigens. The strategy of tumor cell damage to signaling pathways in immune cells involves the synthesis of immunosuppressive cytokines, which can reduce the expression of CD28, 4-1BB/4-1BBL, CD3ζ, and other costimulating factors [19]. Tumor cells are also able to reduce the synthesis of anti-apoptotic proteins in immune cells and cause apoptosis of, for example, T-lymphocytes through a caspase-dependent mechanism [21,22]. At the same time, they increase the expression of their own anti-apoptotic proteins, such as FLIP (FLICE-Like Inhibitory Protein), Bcl-XL (B-cell lymphoma-extra large), and other proteins. They also express deactivated death receptors and reduce the expression of ligands for some chemokine receptors on the surface of immune cells. [19,20].

## 3. Genetic Modifications in CAR Cell Therapy

Since the development of the first anti-tumor cell therapies, there has been a need to improve their effectiveness in killing tumor cells, eliminating them from the body and increasing the lifespan of CAR cells, making cellular therapies safer for cancer patients.

### 3.1. Strategies for Modifying the Chimeric Antigen Receptor

CAR, or chimeric antigen receptor, is a genetically engineered protein molecule that recognizes and binds to a specific antigen. It enhances the effector functions and survival of immune cells that carry it. The CAR molecule consists of four main domains: an antigen-binding domain, hinge domain, transmembrane domain and signaling domain [23,24].

The strategy for constructing the antigen-binding domain involves two approaches: one based on the sequence of an antibody specific for a particular ligand associated with a specific type of tumor and another based on the sequence of a conventional NK cell receptor. In the former case, the antigen recognition CAR domain is composed of sequences derived from the light and heavy chains of a monoclonal antibody, also known as a single-chain variable fragment (scFv), which can be specific for either a particular protein that the antibody binds to or a universal binding protein such as avidin [24,25]. In the second case, the antigen recognition domain of the CAR is represented by a receptor sequence that is normally found on the membrane of NK cells and is specific to certain ligands normally absent on healthy cells in the body, or are present at very low levels. An example of such a receptor is the NKG2D, which binds to MICA, MICB, and ULBP proteins [26]. NKG2DLs are stress-induced proteins that are present on the surface of cancer cells and are found in tumors of the testicles, ovaries, glioblastoma, lung, gastric, colorectal, head and neck, liver, pancreas, kidneys, bladder, prostate, and melanomas [27]. The absence of these proteins from normal cells helps to avoid an autoimmune response. Tumor progression may lead to decreased levels of NKGDLs, making it harder for NK cells to recognize and eliminate tumor cells. However, there are many other activating receptors on NK cells that can still recognize and destroy tumors [28,29]. scFv is the most popular choice for the antigen recognition CAR domain. Antibody sequences are connected in scFv using a linker. With proper length selection, the linker stabilizes the scFv structure [30]. The Whitlow 218 linker sequence is the one that is most often used (GSTSGSGKPGSGEGSTKG) [30,31,32,33]. The best choice for the antigen recognition domain is the antibody sequence specific to the antigen, which is predominantly expressed on tumor cells and rarely found in normal cells. This reduces the toxicity of CAR therapies against normal body cells. In cases where the antigen is expressed in normal cells and increased in tumor cells, a strategy has been developed to reduce the affinity of scFv, so that CAR-bearing cells will not bind to normal cells but mainly react with tumor cells [23,32,34]. As an example of using scFv as the basis for the antigen recognition domain, CAR-NK cells targeting multiple myeloma can be considered. In clinical studies, BCMA (B-cell maturation antigen) is the most popular target antigen for multiple myeloma, and the basis of the antiBCMA-CAR-NK cell’s antigen recognition domain is the sequence of a monoclonal antibody that targets BCMA. However, this antigen target has some disadvantages. For instance, BCMA expression is specific to human B-lineage cells, so using antiBCMA-CAR-NK cells to treat multiple myeloma could potentially eliminate the patient’s B-cell immunity [26,32,33,35]. Studies are also underway to investigate the therapeutic potential of antiBCMA/GPRC5D-CAR-NK and antiCD70-CAR-NK cell therapies for multiple myeloma (NCT06594211, NCT06696846). In addition, screening of multiple myeloma surface antigens has identified other promising targets, including CD28 (Tp44), CD48 (BLAST), CD74 (CLIP), CD138 (SDC1), CD229 (SLAMF3), CD319 (SLAMF7), CCR10 (GPR2), TAC1 (NPK), TXNDC11, SLC1A5 (AAAT) [36]. 

The CAR hinge domain acts as a link between the antigen recognition and transmembrane domains, and by selecting an appropriate length, it helps to stabilize the overall conformation of the receptor. This, in turn, promotes the formation of a stable immune synapse between the CAR-NK cell and a tumor cell. The hinge domain is often based on extracellular domains of CD8a and CD28, as well as the IgG sequence. The CD28 and CD8a-based sequences have the ability to form dimers, leading to increased activation of the receptor, while the IgG-derived sequence provides flexibility to the receptor structure [30,33,37,38].

The transmembrane domain binds the extracellular part of the receptor to the intracellular signaling domain, and it is able to influence the stability and activation of CAR. This domain is often represented by sequences from CD3ζ, CD8, CD28, NKG2D, DNAM-1, and 2B4 molecules. The TM sequence derived from CD28 can lead to better receptor activation, but it also increases the risk of tonic signaling and hyperactivation in CAR-expressing cells. TM sequences derived from NK cell receptors can enhance the cytotoxic activity of CAR-expressing cells [24,30,39].

The intracellular CAR domain plays a crucial role in the activation of immune cells. Based on their design complexity, chimeric antigen receptors can be classified into five generations. The first-generation CAR includes only the CD3ζ signal sequence in its intracellular domain. Studies have shown that T cells expressing these CARs exhibit low antitumor activity. Adding a costimulatory domain to the intracellular sequence of CAR resulted in the development of the second-generation receptors, where the intracellular domain now consists of both CD3ζ and a costimulatory sequence. The third generation of CAR features the addition of two costimulatory domains to the activating CD3ζ sequence. Commonly used costimulatory domains include CD28, ICOS, 4-1BB, OX40, 2B4, DAP10, and DAP12 [23,26,32,39,40,41]. CD3ζ domains, in combination with CD28 or 4-1BB, are most commonly used in the creation of second-generation CAR-T lymphocytes. CD3ζ, CD28, and 4-1BB domains, on the other hand, are used to create third-generation CAR-T [30,33,42,43]. All CAR-cell therapies approved by the FDA by 2025 contain a second-generation CAR with a CD3ζ signaling domain and one of two co-stimulatory domains: CD28 or 4-1BB (see Table 1). For CAR-NK cells, co-stimulatory domains such as 2B4, DAP10, and DAP12 have been shown to be the optimal choice [44].

It has been found that the CD28 costimulatory domain causes a faster and more potent cytotoxic effect. In contrast, 4-1BB induces a weaker but time-persistent cytotoxic activity and promotes the generation of CAR-NK and CAR-T memory cells [26,32]. In addition to activating and costimulatory domains, the fourth-generation CAR includes sequences that regulate the activity of other cells in the immune system. These sequences encode various cytokines, chemokines, and their receptors [23,33,40]. The fifth-generation CAR includes an additional costimulatory domain, along with a binding motif for certain transcription factors. This further regulates the antitumor response of CAR cells and cells of the immune system [33,41,42].

Currently, research is conducted to improve the effectiveness of CAR cell therapy. The suppression of the expression of the target antigen in tumor cells leads to their escape from CAR-NK cells. In order to solve this problem, researchers are developing CAR-NK cells with dual specificity. This approach makes it possible to increase the probability of interaction between CAR-NK cells and tumor targets. An example is antiPD-L1/antiMICA/MICB-CAR-NK92 cells, which are constructed based on the sequences of the PD-L1 antibody and NKG2D, an activating NK cell receptor [45]. Additionally, by modifying NK cells from the peripheral blood of donors with the 158V/V polymorphism of the FCGR3A gene, antiBCMA/antiGPRC5D-CAR-NK cells were generated [46]. Another example is antiCD19/antiBCMA-CAR-NK cells derived from iPSC, which also express genes encoding IL-2RF for improved proliferative activity and tEGFR to ensure the safety of the CAR-NK cell [47]. CAR cells that target two different antigens are being investigated in clinical trials, including NCT04723914, NCT03931720, and NCT03941457. These biCAR cells show better antitumor activity and are more specific to target cells [48]. This not only helps to address the problem of trogocytosis and self-reactivity in CAR cells, but it also enhances the functional activity of these cells [49]. Modern CAR design strategies aim to select genetic modifications that simultaneously increase the safety and antitumor activity of CAR cells (NCT05182073).

### 3.2. Enhancing NK Cell Cytotoxicity Using the High-Affinity, Non-Cleavable CD16 Receptor with Enhanced Functional Activity

The cytolytic activity of NK cells is directly correlated with the sum of the reactions of their activating and inhibitory receptors. However, a signal from one type of activating receptor alone is not enough to activate NK cells, as additional costimulation from other types of receptors is required. This prevents autoreactivity towards normal body cells. However, there is one exception: the CD16a activating receptor (FcγRIIIa) that mediates ADCC and can directly and independently activate NK cells by binding to the Fc fragments of antibodies on opsonized target cells [50,51]. The ability of CD16a to activate NK cells independent of other activating receptors has led to its recognition as the most promising candidate for cellular immunotherapy. CD16 is a member of the Fcγ receptor (FcγR) family and belongs to the FcγRIII group. It exists in two isoforms: CD16a (FcγRIIIa), which is expressed in NK cells, and CD16b (FcγRIIB), which is found on neutrophils [52,53,54]. CD16a binds to homo- or heterodimeric adapter protein complexes CD3ζ and FcRγ, which both contain ITAM (Immunoreceptor Tyrosine-based Activation Motif) motifs [52,55,56,57]. After CD16a binds to the Fc fragment of the antibody, Src family protein kinases phosphorylate the tyrosine residue in the ITAM motifs of CD3ζ and FcR proteins. This creates a binding site for the phosphorylated ITAM fragment with the SYK (Spleen Tyrosine Kinase) and ZAP70 (Zeta-chain-Associated Protein kinase 70) proteins. This leads to the double phosphorylation of the SLP76 (SH2 domain-containing Leukocyte Protein of 76kDa) protein, which ensures phosphorylation of and interaction with two VAV1 proteins simultaneously. This results in a stronger signal in subsequent molecular cascades, leading to the release of lytic granules from the NK cell into the immunological synapse formed with the antibody-opsonized target cell. Compared to other activating receptors, CD16a activation is the only one that leads to double phosphorylation of SLP76 [50,51,55,57,58].

However, in the context of cellular immunotherapy, there are two factors that limit the potential cytolytic activity of unmodified CD16a: the strength of its binding to the Fc fragment of an antibody and its cleavage.

The affinity of CD16a binding to the Fc fragment of antibodies varies depending on the polymorphism in the *FCGR3A* gene of the CD16a receptor. At the 559th nucleotide position of the cDNA, there is either a G or a T nucleotide, which results in either phenylalanine or valine being present at the 158th amino acid position of the receptor sequence [59]. The CD16a protein with amino acid valine at position 158 (V158) has the highest affinity and stability for interaction with antibodies, while the receptor with phenylalanine at position 158 (F158) has the lowest affinity (taking the signal sequence into account, the position will be 176). The dominant allele in humans is the one with phenylalanine. There is evidence that homozygous patients with the low-affinity variant CD16a-176F/F have a worse prognosis for treatment with therapeutic antibodies than those with the heterozygous CD16a-176V/F or homozygous CD16a-V/V genotypes [59,60,61,62,63,64].

After contact with an antibody on an opsonized target cell, CD16a can trigger the secretion of cytolytic granules (Figure 1). This process is regulated by the protein ADAM17 (disintegrin and metalloprotease 17), a metalloprotease located on the surface of NK cells. ADAM17 “cuts off” the extracellular part of CD16a. In the study, Jing, Y., and colleagues identified three separate cleavage sites in close proximity at P1/P1′ positions alanine195/valine196, valine196/serine197, and threonine198/isoleucine199, revealing a membrane proximal cleavage region in CD16 [65]. This mechanism of CD16a inactivation possibly serves to prevent NK cell autoreactivity against normal cells in the body, but in the context of tumor cell elimination, it plays a negative role and leads to a decrease in the effectiveness of immunotherapy in patients [50,52,58,65,66].

Non-cleavable and high-affinity CD16a has been developed to generate therapeutic genetically modified NK cells. This has been made possible by simultaneously substituting phenylalanine for valine at amino acid position 158 and serine for proline at position 197. The non-cleavable variant of the receptor results from the fact that proline has a more rigid, cyclic structure compared to serine. In preclinical studies, the positive impact of these modifications on tumor cell elimination has been demonstrated both in vitro and in vivo, compared to the effects of antibodies alone and unmodified NK cells [50,58,62,63]. To date, clinical trials have started to evaluate the potential of high-affinity, non-cleavable CD16a for the treatment of cancer patients (NCT06342986, NCT05182073). 

An increase in cytotoxic activity can also be achieved by introducing additional intracellular activation domains into the CD16a structure. The receptor sequences found in NK cells or T lymphocytes can be used for these domains [50]. Intracellular activation domains of CD16a amplify the cytotoxicity signal. Transmembrane adapter protein fragments employed in CAR design can serve as these sequences, including DAP10 (DNAX-Activation Protein of 10 kDa) or DAP12 (DNAX-Activation Protein of 12 kDa), signaling domains of CD3ζ or FCεRIγ, as well as sequences of co-receptors and co-stimulators such as 2B4, 4-1BB, CD28, OX40, and DNAM [39,67].

DAP10 and DAP12 are adapter proteins that bind to receptors and contribute to the transmission of a signal to downstream molecules in signaling cascades. DAP10 is associated with the NKG2D activating receptor and contains a YINM (tyrosine-based signaling) motif, instead of an ITAM [68]. Phosphorylation of YINM activates PI3K kinase or GRB2 adapter protein, leading to the activation of cytotoxicity mechanisms [50]. DAP12 is associated with NKp44, KIR2DS, KIR3DS and NKG2C activating receptors. It contains an ITAM sequence that, upon binding of the receptor to its ligand, is phosphorylated by kinases. This phosphorylation causes the involvement of tyrosine kinases SYK and ZAP70. These tyrosine kinases lead to the activation of downstream cytotoxicity signals and the cytokine secretion [51,69].

CD3ζ is a protein that is part of the TCR of T cells. In NK cells, CD3ζ acts as a signaling adapter protein, typically associated with CD16 in a homodimeric or heterodimeric form with FCεRIγ [52,55,56,57]. CD3ζ and FCεRIγ can also serve as adapter proteins for receptors belonging to the NKp family. Both CD3ζ and FCεRIγ contain ITAM sequences, which transmit a signal for the activation of cellular cytotoxicity through phosphorylation of these sequences. This activation is mediated by CD16 [51,57,58].

### 3.3. Membrane-Bound IL-15/IL-15Rα Complex (IL-15RF)—A Strategy for Improved NK Cell Survival

IL-15 is one of the most significant factors regulating NK cell survival and functional characteristics. It is a cytokine with a molecular weight of 14–15 kDa [70]. It has been found that *IL15*, *IL15RA* or *IL2RB* (encoding the β-subunit of IL-15 receptor-IL-15Rβ) gene knockdown leads to a complete stop in the development of NK cells and their inability to differentiate from the CD56^bright^ to the CD56^dim^ phenotype. This data proves the key role of IL-15 in regulating NK cells at all stages of their development, from early precursors to mature cells [71]. IL-15 is produced by dendritic cells, monocytes, and macrophages, in conjunction with the IL-15Rα receptor subunit. It is presented to T cells and NK cells through transpresentation. When IL-15 binds to its IL-15Rα receptor subunit on the surface of an antigen-presenting cell, it interacts with the β and γ subunits of the IL-2/15R receptor on the target cell’s surface, forming a ligand-receptor complex [70,72,73,74,75]. This leads to the activation of STAT3 and STAT5 transcription factors, as well as the activation of the PI3K/AKT/mTOR and RAS/RAF/MAPK signaling pathways. This results in an immunostimulatory effect on targeted cells [70,72,76,77]. In addition, T cells and NK cells are able to present IL-15 in a cis-manner (Figure 2). Both IL-15 and the IL-15Rα receptor subunit and IL-2/15Rβ and IL-2/15Rγ receptor subunits are expressed in the same cell. As a result, IL-15 interacts with the cell that synthesized it and contributes to its activation and maturation [70,72,76,78]. IL-15 can exist in three forms: as a soluble monomer that is secreted, as a non-cellular IL-15/IL-15Rα complex that is also secreted, and as the membrane-bound IL-15/IL-15Rα form (mbIL-15/IL-15Rα). The most potent effect on immune cell activation is achieved through the mbIL-15/IL-15Rα superagonist [70,72,75].

CAR-NK cell therapy has the potential to revolutionize cancer treatment. The risk of side effects from CAR-T therapy has been significantly reduced, and the therapy is now available not only for hematological tumors, but also for solid tumors. However, there are still some limitations, such as the short lifespan of CAR-NK cells in the body and their reduced viability, especially in the tumor environment, which can lead to a rapid decrease in their ability to kill cancer cells [79,80]. In this context, further modification of CAR-NK cells with the mbIL-15/IL-15Rα superagonist could overcome these limitations and improve their viability after transplantation.

According to research, the administration of secreted soluble IL-15 monomer and an unrelated IL-15/IL-15Rα complex to patients can lead to antitumor stimulation of immune cells and their proliferation. However, this treatment carries the risk of serious side effects, as IL-15 itself has been linked to the development of chronic inflammatory diseases and autoimmune reactions [70,74,78]. Monomeric exogenous IL-15 can cause hypotension, thrombocytopenia, liver damage, and sepsis in cancer patients. Prolonged exposure to IL-15 may lead to hyperactivation of CAR cells and their premature depletion. Experimental studies in mice have found that the administration of IL-15 causes neutropenia and lymphocytic leukemia. Additionally, the administration of the IL-15/IL-15Rα complex has been linked to hypothermia and splenomegaly in these animals [70].

It is important to note that IL-15, when combined with IL-15Rα, will produce a more effective response than IL-15 without proper presentation. The creation of the mbIL-15/IL15-Rα complex is made possible through the artificial “binding” of IL-15 to IL-15Rα via a linker sequence [73,81]. Five domains are identified in the structure of IL-15Rα: the sushi domain, which IL-15 binds to with high affinity; the linker domain; the domain with a high proline/threonine content; the transmembrane domain; and the cytoplasmic domain. It was found that the IL-15 complex, which is connected to the sushi domain of IL-15Rα by a linker sequence (IL-15/sushiIL-15Rα), retains the same immunostimulatory effect as the entire IL-15/IL-15Rα complex [70,73,77,81,82]. Several studies have demonstrated the beneficial role of mbIL-15/IL-15Rα in CAR-T and CAR-NK therapy of hematological and solid tumors. Compared to antiCD19-CAR-T cells, antiCD19-mbIL15/IL15Rα-CAR-T cells exhibited increased antitumor activity and proliferation, leading to improved survival rates in experimental mice [83]. In another study, researchers compared anti-MSLN-CAR-T cells with anti-MSLN-mbIL15/IL15Rα-CAR-T cells. They found that the latter had greater antitumor cytotoxicity in vivo, were characterized by a higher proliferation rate, and could survive longer in vitro without IL-2 [84]. However, in the study of antiCD19-mbIL15/IL15Rα-CAR-T cells, some adverse effects of this therapy were also observed. Specifically, mice treated with these cells developed splenomegaly and abnormal proliferation of T cell clones. These effects were not seen when conventional anti-CD19 CAR-T cells were administered to mice [83].

Regarding CAR-NK cells, it was found in vitro that antiCD19-mbIL15/IL15Ra-CAR-NK cells, compared to antiCD19-CAR-NK cells, were viable for 21 days of cultivation without the presence of IL-2, while conventional antiCD19-CAR-NK cells proliferated up to 12 days of cultivation. In vivo, these cells had greater antitumor activity and increased the survival rates of experimental mice [72]. It was also noted that the secreted form of IL-15 can maintain the persistence of CD123/BCMA-CAR-NK cells in vivo for 40–50 days [30].

CAR-NK cells, supplemented with mbIL-15/IL-15Rα, have shown excellent results in preclinical studies, outperforming conventional CAR-NK cells. Clinical trials are currently underway to investigate the effect of mbIL15/IL15Rα-CAR-NK cells on various tumor types in cancer patients (NCT06652243, NCT06342986, NCT05182073).

### 3.4. Immune Checkpoint Inhibition as a Strategy to Increase Cytotoxicity and NK Cell Survival

The modern design of CAR-NK cells is based on three main approaches: increasing antitumor cytotoxicity;increasing the lifespan of cells and the time of their persistence in the patient’s body;increasing their safety for the patient.

To achieve the best results in increasing antitumor cytotoxicity and lifespan, it is possible to suppress the expression of certain genes in NK cells. The choice of the right target depends on several factors. For example, some immune checkpoints can be knocked out or knocked down, through which the tumor suppresses the functional activity of CAR-NK cells. One such example is the NKG2A receptor [85]. And by using the therapeutic antibody daratumumab, it is possible to increase the effectiveness of CAR-NK cell immunotherapy. Daratumumab is an FDA-approved therapeutic monoclonal antibody against CD38 [86,87]. To prevent the fratricide of CAR-NK cells during daratumumab therapy, the CD38 gene knockout is made in addition to the CAR in NK cells. At least one clinical trial has already been launched that examines the effectiveness of combining CAR-NK cells with a CD38 knockout and daratumumab (NCT05182073) [88]. 

AHR (Aryl Hydrocarbon Receptor) is a transcription factor that plays a dual role in the development and maintenance of normal NK cell function. It regulates the expression of a large number of genes, ensuring the formation of a population of CD56^bright^ NK cells through the action of the STAT3 transcription factor. However, AHR agonists produced in the tumor microenvironment can reduce the antitumor cytotoxicity of NK cells and activate the expression of genes responsible for oxidative stress [89,90]. In vitro tests have shown that knocking out 4 genes-*AHR*, *CISH*, *TIGIT*, and *PDCD1*-does not lead to increased antitumor cytotoxicity compared to the combined knockout of only 3 genes-*CISH*, *TIGIT* and *PDCD1*. However, compared to a control group of NK cells with no genetic modifications, both groups of NK cells that had the knockout of the *AHR* and *TIGIT* and *PDCD1* genes and a group with an additional knockout of the *CISH* gene (4 genes in total) showed increased antitumor cytotoxicity [91].

CISH (cytokine-inducible SH2-containing protein) is a protein that belongs to the SOCS (Suppressor of cytokine signaling) family. It is a negative regulator of the STAT5 signaling pathway, which is triggered by IL-15 in NK cells [92,93]. By binding to phosphorylated tyrosine in the intracellular domains of cytokine receptors, CISH prevents further signal transmission by disrupting the signaling cascade. It can also direct the proteins that interact with it towards proteasome degradation, as it interacts with the E3 ubiquitin ligase [93]. A correlation was found between the levels of CISH-positive NK cells, IL-10, and GRP78 and the development of ovarian cancer. This suggests that CISH can be considered as a marker of NK cell exhaustion [94]. Many studies have also noted the positive role of the *CISH* gene knockout in NK cells. This leads to the stability of the JAK-STAT signaling pathway when interacting with IL-15 and, as a result, better survival and longer persistence of NK cells, as well as better antitumor cytotoxic activity in vitro and in vivo. It has been noted that NK cells with *CISH* knockout inhibit tumor metastasis in mouse models in vivo. Taken together, the available data suggest that the *CISH* gene is an attractive target for knockout/knockdown in NK cells [91,93]. 

Important immune checkpoints of NK cells are TIGIT and CD96 receptors. The ligand for these receptors is the CD155 protein, which is expressed at higher levels in various types of malignant tumors [95,96,97,98]. TIGIT has a stronger affinity for CD155, followed by CD96 in terms of binding strength. Since TIGIT contains only the inhibitory ITIM motif, it has a stronger inhibitory effect on the activity of NK cells compared to CD96, which carries not only ITIM, but also the YXXM motif, which under certain conditions can lead to activation of the cell. Increased expression of TIGIT in NK cells that infiltrate tumors has been linked to an increased risk of metastasis and a poor prognosis [99,100]. In studies, *TIGIT* and *CD96* knockout has led to an increase in the cytotoxicity of NK cells and a suppression of tumor metastasis. Harjunpää et al. conducted a study on mice and compared a group with *PDCD1* (which encodes PD-1) and *CD96* knockout to a group with both *TIGIT* and *CD96* knockout. The results showed that the removal of *PDCD1* and *CD96* significantly contributes to the suppression of tumor growth, leading to complete remission [97]. The results of another study demonstrated better antitumor cytotoxicity of natural killer (NK) cells with simultaneous knockout of three genes: *TIGIT*, *PDCD1*, and *CISH*, compared to knockout of two genes, *TIGIT* and *PDCD1*. This was also compared to the knockout of three other genes, *TIGIT*, *PDCD1* and *AHR*, as well as the knockout of four genes, *TIGIT*, *PDCD1*, *AHR* and *CISH*. In the same study, mice that received anti-CD19-CAR-NK cells expressing the soluble form of IL-15 and with the knockout of *TIGIT*, PDCD1, and *CISH* had better indicators of antitumor activity and longer survival compared to the other three groups: anti-CD19-CAR-NK, anti-CD19-CAR-NK with IL-15 expression, and anti-CD19 CAR with only *TIGIT*, *PDCD1* and *CISH* knockout [91].

Much attention in modern research has been focused on the NK cell inhibitory receptor NKG2A (CD159a). This receptor’s ligand is the HLA-E molecule, which is expressed in high numbers by some tumor cells in order to evade the immune system’s surveillance. NKG2A is encoded by the *KLRC1* gene, and in NK cells and CD8+ T lymphocytes, it is localized on the cell membrane as a heterodimer with the CD94 receptor [101,102,103]. The limited amount of HLA-E on the surface of normal cells helps them avoid an autoimmune reaction; however, in tumor cells, the cytotoxicity of NK cells is inhibited due to an increase in HLA-E. This increase is associated with a poor survival prognosis. In addition, NK cells that infiltrate tumors express NKG2A at much higher levels than normal NK cells, which is due to the influence of factors in the tumor microenvironment and various cytokines such as IL-21, IL-12, IL-10, and TGF-β (Transforming Growth Factor beta) [101,102,104]. When NKG2A binds to its ligand, two of its intracellular ITIM domains are phosphorylated. This attracts SHP phosphatases, triggering a cascade that suppresses the cytotoxicity of NK cells [101,105]. In many experimental studies, it has been noted that inhibiting the functional activity of NKG2A increases the antitumor cytotoxicity of NK cells. It was shown that *NKG2A* gene knockout with the CRISPR/Cas9 system appears to be much more effective than the use of therapeutic antibodies against NKG2A [101]. However, a recent study has shown that NKG2A helps maintain the ability of NK cells to proliferate and protects them from apoptosis. Therefore, it is important to treat *NKG2A* gene knockout in NK cells with caution [103].

PD-1 is a protein encoded by the *PDCD1* gene, which is well known for its immune-suppressive function in T lymphocytes. It binds to its ligand PD-L1, which is highly expressed by certain types of tumor cells. This causes the development of functional anergy in T-lymphocytes [106,107]. The immunosuppressive effect of PD-1 is due to the presence of an inhibitory ITIM domain. This domain becomes phosphorylated when the receptor interacts with its ligand. This interaction attracts the tyrosine phosphatase SHP-2, which dephosphorylates the CD3 coreceptor and the ZAP70 tyrosine kinase associated with TCR in T lymphocytes. This prevents transmission of an activating signal from TCR [107,108]. There is conflicting data on the expression of PD-1 on NK cells. It is generally believed that NK cells from healthy individuals do not express PD-1. However, studies have shown that patients with certain tumor types and certain viral infections may have NK cells that express PD-1. These conditions include multiple myeloma, renal cell carcinoma, Kaposi’s sarcoma, ovarian carcinoma, Hodgkin’s lymphoma, digestive system tumors, non-small cell lung cancer, and patients with cytomegalovirus or HIV infection [107]. It is likely that PD-1-positive NK cells may also be found in the development of other infectious and tumor diseases as well [109]. There is also evidence that NK cells can acquire PD-1 from the surface of tumor cells via a process called trogocytosis [110]. At the same time, PD-1 expression by NK cells themselves is mainly observed in a highly differentiated population of CD56^dim^, CD57-positive NK cells. These PD-1-positive NK cells also have a low level of expression of activating receptors, such as NKG2D, NKp30, NKp46, and DNAM-1, and reduced secretion of IFNγ, TNF-α, and granzymes [111,112]. The FDA has approved PD-1 immune checkpoint inhibitors, which are a type of monoclonal antibody, for the treatment of several types of tumors [113,114].

LAG3 (Lymphocyte Activation Gene 3, also known as CD223) is a protein expressed by NK and T cells. Its biological function is to inhibit the expression of IFNγ by inhibiting the STAT1/IFNγ molecular pathway. It also disrupts the process of glycolysis by inhibiting the PI3K/AKT/mTOR pathway and inhibits T cell activation by inhibiting TCR [115,116]. LAG3 ligands include MHC class II molecules, Galectin-3, L-selectin, α-synuclein, FGL3, and TCR/CD3 complex [117,118,119]. Increased expression of LAG3 on NK and T cells has been observed in the development of viral diseases. In particular, LAG3-high NK cells are found in HIV-infected individuals, as well as in tumor diseases, which is associated with a poor survival prognosis [115,116,120]. The expression of LAG3, as well as other immune checkpoint molecules, is particularly strongly increased in tumor-infiltrating lymphocytes. In follicular lymphoma, an increased co-expression of LAG3 with TIM3 and PD-1 on T cells is observed [116]. It has been reported that when LAG3 is inhibited, the expression of the PD-1 protein gene increases. This leads to the conclusion that there may be a potential advantage in combining the blockade of multiple immune checkpoints. Preclinical studies and clinical trials have confirmed the positive effect of LAG3 inhibition on lymphocytes [118,121,122]. 

Additional data on the listed immune checkpoints and some other relevant information are presented in Table 2. Two important points should be noted: firstly, drugs currently approved by the FDA that are immune checkpoint inhibitors are monoclonal antibodies; and secondly, none of the CAR-cell therapies approved by 2025 involved the knockout of immune checkpoints.

### 3.5. Safety Switches

CAR cell immunotherapy, despite its many advantages, does have some limitations. One such limitation is the potential toxicity of the CAR-T cell therapies, which has been demonstrated in clinical trials, and the potential toxicity of CAR-NK cell preparations. Another limitation is the risk of insertion mutagenesis and oncogenic transformation during the process of transducing immune cells with CAR-carrying vectors [152]. To ensure patient safety, one solution is to completely eliminate CAR cells using protein switch molecules that induce apoptosis. There are several artificial approaches to induce apoptosis in cells, including the use of chemicals that induce dimerization of pro-apoptotic proteins, the interaction between ganciclovir and HSV-TK (herpes simplex virus thymidine kinase), and antibody-dependent cellular cytotoxicity-based methods [153].

The first approach is based on the use of a modified version of caspase 9, known as inducible caspase 9 (iCasp9), which can be activated by homodimerization after the introduction of a specific chemical inducer. Caspase 9 is a key enzyme in the initiation of apoptosis, a process that leads to programmed cell death. It has a proteolytic function that triggers the mitochondrial (intrinsic) apoptosis pathway [154]. The iCasp9 gene contains the sequence for human caspase 9, without the natural activation domain CARD (Caspase Activation and Recruitment Domains), and the sequence for the human protein FKBP. When this sequence interacts with FK1012, which is a molecule consisting of two tacrolimus molecules (FK506), it can form a dimer. Chemical inducers of this dimerization include AP20187 and AP1903 (or rimiducide) drugs. These inducers have a molecular structure that is almost identical to that of FK1012. When AP1903 or AP20187 are administered, they interact with iCasp9 and cause it to form a dimer, triggering apoptosis in the cell [155]. The advantages of this approach include high efficiency in eliminating target cells (more than 90%) and a fast rate of cell removal (within half an hour). Another benefit is the selective effect of the drugs administered, which interact only with the specific FKBP sequence. However, there are some disadvantages to this method. AP20187 and AP1903 have not yet been approved for clinical use, so they are not available for widespread use. It is possible that a large number of clinical trials showing positive results will soon change this situation (NCT03056339, NCT05020015, NCT00710892, NCT01494103, ACTRN12614000290695) [152,156,157]. 

The second approach is based on the interaction between ganciclovir and the herpes simplex virus type 1 thymidine kinase (HSV-TK). When genetically modified chimeric antigen receptor (CAR) cells are produced, the HSV-TK gene is introduced into them. This gene is not normally present in human cells, so when patients are treated with ganciclovir, it selectively eliminates HSV-TK-expressing CAR cells. Upon HSV-TK action, ganciclovir converts into a toxic metabolite that causes apoptosis of CAR cells. Clinical trials have shown promising results for this approach in eliminating HSV-TK-positive cells. However, HSV-TK may also cause an immune response in the body. Another limitation of this method is the relatively long time needed to eliminate the HSV-TK cells, approximately three days compared to the use of iCasp9, which takes only half an hour [153,157,158]. 

The third approach is based on the ability of NK cells to induce antibody-dependent cell death of targeted cells after interaction with antibodies bound to genetically modified proteins on the surface of these cells. To address this issue, genes for truncated epidermal growth factor receptor (tEGFR) (NCT03084380, NCT05141253, NCT05166070) or RQR8, a protein consisting of two epitopes (CD20 and CD34) (NCT05211557), are introduced into cells. Due to the deletion of intracellular signaling domains, these artificial proteins are unable to trigger signaling cascades in cells. However, the preservation of antibody binding epitopes in tEGFR and RQR8 allows them to be used as markers for transduced cells. This makes it possible to not only specifically eliminate “labeled” cells, but also conduct cell sorting. The advantages of this approach include the opportunity to use clinically approved antibodies such as cetuximab for tEGFR or rituximab for the CD20 epitope in RQR8. The constructs introduced into cells do not cause immunogenicity, and there is a high rate and efficiency of elimination, with more than 80% of cells being eliminated. The disadvantages of this approach include, firstly, its potential limited effectiveness in patients who have undergone lymphodepletion therapy prior to CAR-T treatment, as it may reduce the functional activity of NK cells. Secondly, there is a risk of cross-reactivity known as off-target effects of antibodies with EGFR or CD20 on healthy cells of the patient, leading to their elimination [153,156,158,159].

## 4. Modern Clinical Trials of CAR-NK Cell Products

Genetically modified NK cells are a promising tool for cancer immunotherapy. The interest in developing this area, the prospects of the field, and the number of clinical trials registered on ClinicalTrials.gov all correlate with each other. As of September 2025, more than 100 clinical trials had been registered investigating the effectiveness of genetically modified NK cell therapy in eliminating various types of tumors. In most cases, CAR-NK cells are the subject of research, and these are primarily first-phase trials. A small number of trials are in the second phase. As with CAR-T, hematological malignancies such as B-cell lymphoma and acute myeloid leukemia are still the primary focus for exploring the therapeutic anticancer potential of CAR-NK cells. However, a significant number of clinical trials of CAR-NK cells are focused on assessing their effectiveness against various types of solid tumors, including non-small cell lung cancer, small cell lung cancer, colorectal cancer, and pancreatic cancer, among others. These trials are all registered on ClinicalTrials.gov. The trials involving genetically modified NK cells can be roughly divided into several categories (see Table 3): 

A group of clinical trials using the “classic” CAR structure. In this type of CAR, the antigen-recognition domain is derived from an antibody that is specific to a particular target. This domain contains a single-chain variable fragment (scFv), which is based on the variable light (VL) and variable heavy (VH) antibody chains. These types of CAR target a specific antigen (NCT06454890, NCT06690827);

A clinical trial group using a non-standard CAR structure. In this instance, the antigen-recognition domain is represented by the sequence of an activating NK cell receptor. For example, the NKG2D fragment is used more frequently than others. This type of CAR is versatile and can be applied to various nosologies (NCT05247957, NCT05213195).

A group of trials using specific approaches to the design of CAR-NK cells:

Bispecific CAR (biCAR) aimed at two antigens simultaneously (NCT03931720; NCT06594211).

NK cell modification that involves the simultaneous expression of several protein structures. For example, a “classic” CAR is directed against a specific antigen, while additional proteins are synthesized. This approach was used in the NCT05987696 trial, where the auxiliary protein was chemokine CCL1, which attracts other immune cells to the site of infection.

NK cells without CAR, but with other genetic modifications, such as gene knockouts (NCT04991870).

Patients with B-cell non-Hodgkin’s lymphoma, B-lymphoblastic leukemia, multiple myeloma, acute myeloid leukemia, and T-cell leukemia/lymphoma are usually selected for treatment of hematological malignancies with CAR-NK cells. CD19, CD22, BCMA, CD70, CD123, and CD33 are the most common targets for these treatments (Table 4). Clinical trials of CAR-NK therapy for solid tumors are less common. Ovarian carcinoma, pancreatic cancer, endometrial carcinoma, and hepatocellular carcinoma are the solid tumors most often studied in CAR-NK cell therapy. MUC1, CD70, TROP2, MICA/B, GPC3, and claudins are common targets for designing CAR-NK cells for solid tumors.

CAR-NK cell therapy is a promising approach to the treatment of cancer. Its development started with high-dose therapy using autologous NK cells and the creation of “classic” CAR-NK cells. Today, it continues with the development of fourth-generation CAR-NK cells that are genetically modified with not only CAR but also other protein constructs. These cells have several functions: they exhibit strong antitumor toxicity, activate the immune system of the patient, maintain their own functionality, and are safe for patients. The analysis of genetic modifications to CAR-NK cells allows identifying several approaches that can increase the antitumor activity of these cells:-Firstly, choosing the right tumor target. Ideally, this target should not be expressed by normal cells. However, in reality, the best option would be a target that is characterized by low expression in healthy cells and high expression in tumor cells.-Secondly, the correct design of the CAR, the choice of optimal intracellular domains, depends on the tasks set.-Thirdly, the introduction of high-affinity, non-cleavable CD16 molecules with intracellular domains into CAR-NK cells will allow the use of therapeutic antibodies in combination with CAR-NK therapy for the treatment of patients. This will provide antitumor activity through ADCC (antibody-dependent cell-mediated cytotoxicity) due to the strong intracellular signal generated by the CD16 molecule (Figure 3).

Better proliferative activity and survival can be achieved by introducing IL-15, bound to its IL-15Rα receptor, into CAR-NK cells. Due to its cis presentation, IL-15/IL-15Rα ensures the continuous self-activation of CAR-NK cells.

Inhibition of immune checkpoints simultaneously performs two functions: it enhances antitumor cytotoxicity and improves the survival of CAR-NK cells, as tumor cells are able to negatively regulate the functional activity of these cells through these checkpoints.

The safety of CAR-NK cells for patients is ensured by incorporating genetic sequences known as “switches” into these cells. These switches allow for the rapid elimination of CAR-NK cells when necessary.

## 5. Optimization of Antitumor CAR-NK Therapy

NK cells, when at rest, have a low demand for glucose, and as a result, the processes of glycolysis and oxidative phosphorylation are slowed down in them. However, when they are activated by cytokines, the glucose consumption level increases significantly, intensifying the processes of glycolysis and oxidative phosphorylation by, in particular, increasing the synthesis of the GLUT1 protein (glucose transporter member 1) [185]. As a result, the effector activity of NK cells increases. However, when NK cells enter the tumor microenvironment (TME), they encounter its toxic effects due to the presence of cytotoxic metabolites, hypoxia, and exposure to tumor-associated immune cells (MDSC, M2 macrophages), and other factors. There is competition for nutrients, as tumor cells consume large amounts of glucose and amino acids. Solid tumor TME is characterized by a reduced concentration of arginine and leucine, as well as the presence of the tryptophan metabolite L-kynurenine and nitrogen oxides, which are products of arginine catabolism. All of the above-mentioned factors lead to changes in the metabolic processes of NK cells, which in turn lead to a deterioration of their effector functions and proliferative activity. This also results in suppression of the ADCC mechanism and a decrease in IFN-γ synthesis. High levels of adenosine, HIF-1α, and lactate cause a decrease in oxidative phosphorylation and glycolysis, as well as a reduction in the expression of activating receptors such as NKp30, NKp46, and NKG2D. Additionally, it leads to decreased production of IFN-γ and granzyme B and inhibits the activity of the transcription factor SREBP [185]. Thus, it is essential to optimize the metabolic functions of CAR-NK cells in order to better eliminate solid tumor cells. Experimental studies have shown that when mice infected with cytomegalovirus are administered a glycolysis inhibitor, 2-deoxyglucose, a reduction in the cytotoxicity of NK cells and an impaired ability to eliminate the virus from the body are observed. This is due to impaired glycolytic and oxidative phosphorylation processes. Moreover, if IL-15 ALT803 superagonist (IL-15 superagonist ALT-803) is injected into mice along with 2-deoxyglucose, it enhances glycolytic activity, oxidative phosphorylation, and the proliferation of NK cells. By day 10, 100% of the mice survive, unlike those that were not treated with ALT803 [186]. In another study, researchers found that the metabolic properties of CAR-NK cells could be optimized by combining the pre-culturing of NK cells isolated from blood with cytokines and genetically modified feeder cells, such as K562 or 721.221. In particular, when NK cells are cultured in the presence of IL-2, IL-15, and the mbIL-21-721.221 feeder cell line, it is possible to generate antiCD19-CAR-NK cells with high antitumor activity both in vitro and in vivo. This is achieved by upregulating the expression of genes involved in amino acid metabolism and glycolysis, such as FN3K (Fructosamine-3-kinase), PEMT (Phosphatidylethanolamine N-methyltransferase), MARS (Methionyl-tRNA synthetase 1), and GPT2 (Glutamic--pyruvic transaminase 2). It has been shown that combined antitumor therapy with CAR-NK cell treatment can promote better metabolic activity in these cells. Standard treatments can lead to a reduction in tumor mass, which in turn reduces the competitive consumption of glucose by tumor cells. This allows CAR-NK cells to consume more glucose, enhancing their antitumor and cytolytic functions [187].

Despite the numerous advantages of CAR-NK cell therapy, there are several challenges associated with this treatment method. One of the main challenges is tumor cells’ immune surveillance avoidance and their inhibitory effect on immune cells after NK cell transplantation. Additionally, the low ability of NK cells to expand in vivo and the effect of the immunosuppressive tumor microenvironment contribute to the shift in the balance of NK cell surface receptors towards inhibitory receptors over activating ones.

Today, researchers are considering various approaches to combination therapy in order to overcome these limitations. Among them is the use of immune checkpoint inhibitors (atezolizumab, nivolumab, and pembrolizumab), cisplatin, radiation therapy (brachytherapy using 125I Seed), tyrosine kinase inhibitors (regorafenib, sorafenib, and cabozantinib), proteasome inhibitors (bortezomib), STING pathway agonists (cGAMP), oncolytic viruses, monoclonal antibodies, and photothermal therapy [188,189,190]. Many approaches are currently being investigated, both in vitro and in vivo. Clinical trials have also started. Thus, in 2021, a phase II clinical trial began to evaluate the effectiveness of allogeneic gamma-irradiated anti-PD-L1 CAR-NK cells in combination with pembrolizumab, a PD-1 inhibitor and N-803, an IL-15 receptor agonist, for the treatment of patients with recurrent or metastatic gastric or head and neck tumors (NCT04847466). Despite the fact that the results of the study have not yet been published, the effectiveness of combining various types of CAR-NK cells with PD-1 inhibitors has been repeatedly demonstrated in preclinical settings using in vitro and in vivo models. Monoclonal antibodies to PD-1 have been shown to enhance the antitumor effects of NK cells [191,192] and increase their cytotoxicity in models of cisplatin-resistant lung cancer [193]. In another exploratory study conducted with heavily pretreated patients with metastatic colorectal cancer, the clinical benefits of combined therapy were observed in a group of three patients who received NKG2D CAR-NK cells in combination with anti-PD-1 treatment. One of the three patients achieved disease stabilization, and two had long-term overall survival of more than 700 days. In the group of patients who did not receive anti-PD-1 treatment, the results were less positive: two patients died at different times after receiving CAR-NK cell injections (the deaths were not related to the treatment), and the third patient was monitored for about 300 days before monitoring was stopped [194]. In another study, an in vitro and in vivo evaluation of the possibility and effectiveness of combining a C021 oncolytic virus, which targets ROR1-positive glioblastoma cells and activates IL-21 secretion in them, with anti-ROR1-CAR-NK cells was carried out [189]. IL-21 is a cytokine that plays a role in the regulation of NK cell functions, including their activation and proliferation. Experiments have demonstrated that IL-21 enhances the activation of NK cells and promotes NK cell secretion of other cytokines such as IFN-γ [195]. The authors of the study demonstrated, in vivo, the ability of the virus to infect neuroblastoma cell lines CHLA-255 and SKNFI. They found that the virus significantly reduced the viability of these cells and increased IL-21 production in surviving cells. This effect was not seen when the virus was added to NK cells. In cytotoxicity experiments, the authors demonstrated that the combination of an oncolytic virus and anti-ROR1-CAR-NK cells significantly enhanced the cytotoxic effect of the latter on SKNFI neuroblastoma cells. Moreover, the combination of oncolytic virus and anti-ROR1-CAR-NK cells significantly increased the survival of NSG mice bearing human neuroblastoma xenografts, demonstrating the effectiveness of this combination in vivo [189]. Thus, the use of oncolytic viruses can simultaneously achieve two goals: enhance the activation and proliferation of CAR-NK cells, and additionally destroy tumor cells.

When choosing treatment options, it is also possible to take into account the individual characteristics of a patient’s tumor. For example, patients with non-small cell lung cancer who have mutations in the gene that encodes EGFR (epidermal growth factor receptor) may be prescribed tyrosine kinase inhibitor (TKI) therapy. However, after TKI therapy, some patients retain cells that are tolerant to the drug, which can subsequently lead to the development of completely therapy-resistant cells. To avoid this, a combination of TKI with anti-EGFR CAR-T or CAR-NK cells that can effectively destroy EGFR-positive tumor cells has been proposed. With prolonged TKI treatment, the number of EGFR molecules on the surface of therapy-resistant cells increases, and combination with anti-EGFR-CAR cells can help target these cells. The combination of TKI with anti-EGFR-CAR-NK cells has been shown to be effective in vitro and in vivo [188].

Therefore, combination therapy takes into account both the individual characteristics of the patient and the tumor, as well as the biological characteristics of CAR-NK cells. This allows for different approaches to improve the effectiveness of antitumor therapy.

## 6. Conclusions

The future of CAR-NK cell therapy lies in the development of a highly effective and versatile “universal” cell. This cell should possess the following features simultaneously:(1)Powerful and versatile antitumor activity(2)Long-lasting presence in the body(3)Impeccable safety profile.

CAR-NK cells have several advantages over CAR-T cells. For example, they do not cause side effects and are less dependent on a single target antigen. In contrast to CAR-T cells, CAR-NK cells maintain the activity of their natural receptors (NKG2D, DNAM-1, and NCRs). This allows them to attack tumors even if the target antigen has been lost, significantly reducing the risk of tumor recurrence. This makes NK cells a promising model for adoptive cellular immunotherapy. Some of their limitations can be easily overcome by a well-selected combination of genetic modifications. To ensure long-term survival of NK cells without external cytokine support, a strategy of internal cytokine production is actively employed, for example, through the expression of membrane-bound IL-15/IL15Ra. To improve safety, cells are modified by introducing “molecular switch” genes. This high antitumor activity is achieved not only through CAR, but also through highly affinity non-cleavable CD16 with intracellular activation domains. Additionally, cells can be modified to resist the immunosuppressive effects of the tumor microenvironment (TME) by making them resistant to TGF-β or hypoxia. Although no CAR-NK drugs have yet been approved by either the FDA or EMA, numerous phase I/II clinical trials have yielded encouraging results, not only for hematological malignancies but also for solid tumors. The variety of NK cell sources and the ability to produce them using a standardized method from iPSCs, as well as the accumulation of positive clinical data, give hope that the first approval of a CAR-NK product may occur in the near future, ushering in a new era of adoptive cell therapy that is safer and more accessible.

## Figures and Tables

**Figure 1 cells-14-01858-f001:**
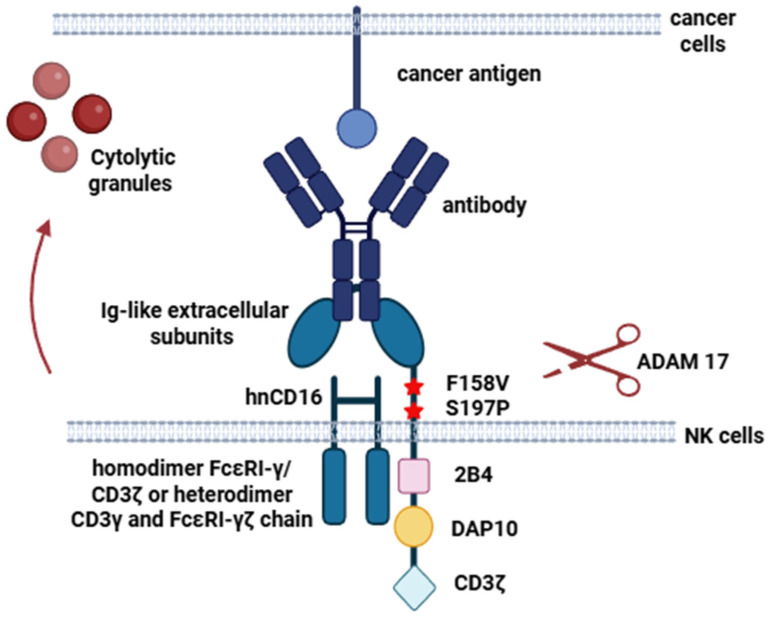
The structure of the high-affinity, non-cleavable CD16 receptor with intracellular domains (created in Biorender).

**Figure 2 cells-14-01858-f002:**
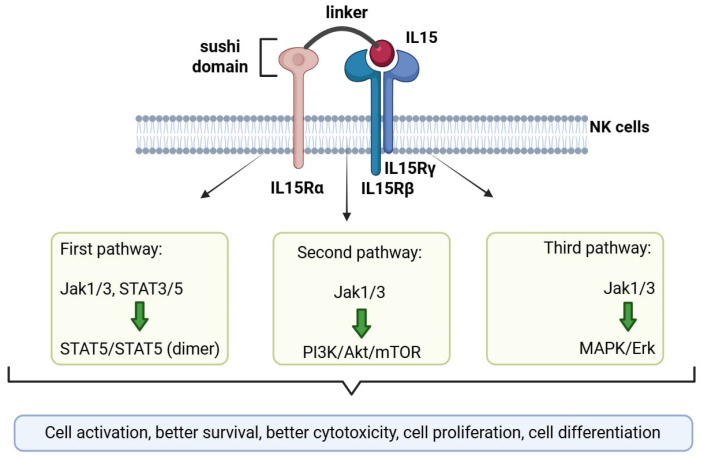
The IL-15/IL-15Rα structure and its cis-presentation to NK cells, as well as the intracellular signaling pathways activated by IL-15, are illustrated (created in Biorender).

**Figure 3 cells-14-01858-f003:**
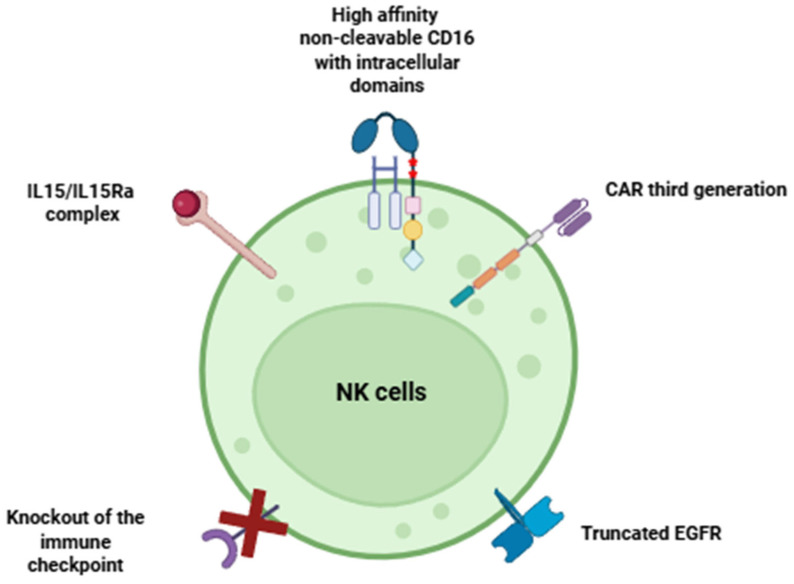
A next-generation CAR-NK cell with high antitumor cytotoxicity, capable of prolonged functional activity and safe for patients (created in Biorender).

**Table 1 cells-14-01858-t001:** FDA-approved CAR-T-cell products.

CAR-T Product	Approval Year *	Target	Intracellular Domains	Tumor Type
Tisagenlecleucel	2018	CD19	4-1BB + CD3ζ	Acute lymphoblastic leukemia, large B-cell lymphoma, follicular lymphoma
Brexucabtagene autoleucel	2020 **	CD19	CD28 + CD3ζ	Acute lymphoblastic leukemia, mantle cell lymphoma
Idecabtagene vicleucel	2021	BCMA	4-1BB + CD3ζ	Multiple myeloma
Axicabtagene ciloleucel	2022	CD19	CD28 + CD3ζ	Large B-cell lymphoma
Lisocabtagene maraleucel	2022 ***	CD19	4-1BB + CD3ζ	Large B-cell lymphoma, follicular lymphoma, chronic lymphocytic leukemia, small lymphocytic lymphoma, mantle cell lymphoma
Ciltacabtagene autoleucel	2022	BCMA	4-1BB + CD3ζ	Multiple myeloma
Obecabtagene autoleucel	2024	CD19	4-1BB + CD3ζ	Acute lymphoblastic leukemia

*—the year is indicated for adults; **—2021 for lymphoblastic leukemia; ***—early approval for B-cell lymphoma, approval in 2024 for other conditions.

**Table 2 cells-14-01858-t002:** NK cell immune checkpoints.

Immune Checkpoint	Cell Specificity (in Descending Order of Gene Expression Levels) *	Intracellular Signaling Pathway	Function	Clinical Trials (NCT)
AHR	Dendritic cells, monocytes, granulocytes, NK cells, T lymphocytes, B lymphocytes	AHR is a transcription factor that is found in the cytoplasm in an inactive state, bound to other proteins. When AHR binds to specific ligands, the complex dissociates, allowing AHR to become activated. Once activated, AHR moves to the nucleus where it interacts with ARNT to form a heterodimer that can affect gene expression. In addition, other proteins may bind to the AHR/ARNT complex through a non-canonical pathway [123,124].	Under normal conditions, it affects the expression of genes involved in cell proliferation and promotes the differentiation of NK cells in a specific direction. In the tumor microenvironment, altered tryptophan metabolism products, such as kynurenine and kynurenic acid, act as ligands for the AHR and suppress the antitumor activity of NK cells by activating genes that lead to oxidative stress, such as *CYP1A*, genes for NADPH oxidases, and cyclooxygenases, as well as by suppressing genes that express antioxidants, such as *NQO1* and *TXN* [90,125].	04069026 (2024); 04999202 (2025); 06874257 (2025)
CISH	T lymphocytes, granulocytes, monocytes, dendritic cells, NK cells, B lymphocytes	Binds to phosphorylated proteins JAK1, JAK3, and STAT5 (JAK-STAT signaling pathway). Binds to the IL-2Rb subunit [126,127].	Inhibits the signals from IL-15 and IL-2 directly. Directs the proteins of the JAK/STAT signaling pathway that are bound to it to proteasomal degradation, attracting an E3 ubiquitin ligase. Inhibition of the JAK/STAT pathway leads to a decrease in the survival of mature NK cells, a disruption in their maturation, and a disruption of the production of perforin and granzyme. The perception of signals from other cytokines is also disrupted, resulting in decreased cytotoxicity [126,127,128].	05566223 (2022); 04426669 (2025)
TIGIT	T lymphocytes, NK cells	TIGIT binds to its main ligand, CD155. Then, the ITT-like domain and ITIM domain become phosphorylated. They bind to the GRB2 protein and phosphatase SHIP1 becomes involved. This leads to the inhibition of the PI3K/AKT and MAPK signaling pathways. Phosphorylation of the ITT-like domain also attracts β-arrestin 2, which binds to the ITT-like domain and also attracts phosphatase SHIP1. SHIP1 disrupts the autoubiquitination of TRAF6, leading to NF-κB suppression [129,130,131].	PI3K/AKT pathway inhibition results in impaired sensitivity of NK cells not only to IL-15, but also to other cytokines. This leads to decreased proliferation of NK cells, suppression of IFNγ and granzyme B synthesis, impaired signal transmission from activating receptors, and impaired polarization of lytic granules. Glycolysis is inhibited. Collectively, the cytotoxicity of NK cells decreases. Inhibition of the MAPK pathway primarily affects the proliferation and differentiation of NK cells. It also disrupts the distribution of perforin B and granzymes in the immunological synapse. Inhibition of the NF-κB signaling pathway disrupts rapid synthesis of pro-inflammatory cytokines such as IFNγ, TNF-α, IL-6, IL-1β, as well as the synthesis of anti-apoptotic proteins. This, in turn, disrupts proliferation of NK cells [129,130,131].	04354246 (2020); 04995523 (2021); 05607563 (2022); 05537051 (2023); 06003621 (2023); 05645692 (2023); 06784947 (2025); 06754501 (2025); 06713798 (2025); 06250036 (2025);
NKG2A/CD94	NK cells T lymphocytes,	It contains two ITIM domains. When the receptor interacts with the HLA-E ligand, tyrosine phosphorylation of the ITIM domain occurs due to the action of SRC and BTK kinases. This triggers the recruitment of phosphatases SHP1 and SHP2, which disrupts the intracellular signaling pathways PI3K/AKT, MAPK, and NF-κB [132,133,134].	The effects of inhibiting the PI3K/AKT, MAPK, and NF-κB pathways are discussed above.	02557516 (2015); 04590963 (2020); 05162755 (2021); 06892223 (2021); 06094296 (2023); 06152523 (2023); 06162572 (2024); 06116136 (2024); 06662669 (2024); 06952010 (2025);
PD-1	T lymphocytes, B lymphocytes. NK-only with the development of certain types of tumors	Contains the ITIM and ITSM domains. Upon binding to the ligand, tyrosine phosphorylation occurs in these domains. This process is associated with the involvement of phosphatases SHP1 and SHP2, as well as the disruption of intracellular signaling pathways, including PI3K/AKT and MAPK, as well as NF-κB [108,135].	The effects of inhibiting the PI3K/AKT, MAPK, and NF-κB pathways are discussed above.	06751901 (2024); 06620822 (2024); 06952010 (2025); 07110103 (2025); 07090707 (2025); 07068763 (2025); 07062484 (2025); 07132528 (2025);
CD96	T lymphocytes, NK cells, B lymphocytes, granulocytes	It contains the ITIM inhibitory domain, which has an inhibitory effect on the cell through the mechanism described above. It also contains the Tyr-XX-MET motif, which increases the functional activity of NK cells after tyrosine phosphorylation, leading to its recognition by the protein subunit p85 of the PI3K/AKT signaling pathway [100,136,137,138].	Modern data is still limited, but it has been shown that CD96, expressed by NK cells, can reduce the production of IFNγ and granzyme B and has a negative effect on NK cell proliferation. This leads to an increase in the expression of anti-inflammatory cytokine IL-10. It is likely that inhibition of NK cells via the ITIM CD96 motif is responsible for these effects. Experimental data confirms the best antitumor activity of NK cells when CD96 is inhibited, suggesting it as a potential therapeutic target for cancer treatment. On the other hand, the presence of the Tyr-XX-MET motif can activate the PI3K/AKT intracellular signaling pathway, favoring both the cytotoxic activity and proliferation of NK cells [100,136,137,138].	03739710 (2019); 04446351 (2020);
LAG3	T lymphocytes, NK cells **	It contains intracellular motifs RRFSALE, KIEELE, EX/EP. The features of LAG3 signal transmission are still poorly characterized. On the one hand, the inhibitory function of LAG3 occurs when the receptor interacts with the ligand and releases its RRFSALE motif from the cell membrane. Next, the cytoplasmic tail is ubiquitinated by ligases of the Cbl family, which leads to an increase in the functional activity of the receptor. Activation of the receptor leads to inhibition of the PI3K/AKT and JAK/STAT signaling pathways. The consequences of inhibiting these molecular pathways are described above [115,139,140,141]	Modern data is still limited. LAG3 has been shown to inhibit the PI3K/AKT and Jak/STAT signaling pathways, with the effects described above. The functional activation of LAG3 results in a decrease in the expression of the cell proliferation activity marker Ki67 [115,139,140,141].	01968109 (2013); 02460224 (2015); 03005782 (2016); 03311412 (2017); 03489369 (2018); 03470922 (2018); 04140500 (2019); 04641871 (2020); 03538028 (2020); 05002569 (2021);
TIM3	Dendritic cells, NK cells, monocytes, T-lymphocytes, granulocytes	It has various effects on NK cells and T lymphocytes. In NK cells, it can promote their activation, while in T lymphocytes, it exhibits a more prominent inhibitory function. The cytoplasmic tail of the protein contains five tyrosine motifs. In the absence of interaction with a ligand, these tyrosine motifs are bound to the Bat3 protein. This activates the catalytic tyrosine kinase LCK, which can lead to the activation of the immune cell. When TIM3 binds to a ligand, such as galectin 9, Bat3 is cleaved and the tyrosine motifs on TIM3 are phosphorylated. It regulates the signaling pathways of NF-κB and MAPK [142,143,144,145].	It has both activating and inhibitory effects on cells, especially NK cells. It disrupts the functioning of the immune synapse. In T lymphocytes, it leads to a decrease in the production of IFNγ, TNF-α, IL-2. In NK cells, it inhibits the CD107a protein, a marker of NK cell degranulation, but it does not affect IFNγ synthesis. The data is varied. In one study, it was proved that despite the fact that TIM3 does not directly affect the synthesis of IFNγ, its inhibition indirectly leads to a more increased synthesis of IFNγ. In another study, if NK cells had been previously treated with cytokines, TIM3 had a stimulating effect on them [142,143,144,145].	02608268 (2015); 02817633 (2016); 03066648 (2017); 03099109 (2017); 03652077 (2018); 04370704 (2020); 04931654 (2021); 04812548 (2021); 05216835 (2022); 05287113 (2022);
IL-1R8	T lymphocytes, NK cells, granulocytes, monocytes, dendritic cells, B-lymphocytes	It contains an intracellular TIR domain with amino acid substitutions that confer inhibitory properties, in contrast to the native TIR. When it binds to the native TIR on interleukin receptors (ILRs, such as IL-1R) and Toll-like receptors (TLRs), the signal from these receptors is inhibited. This is because TIR/IL-1R8 prevents the recruitment of signaling adapter proteins, such as MyD88, TRAM, SARM, TRIF, etc. These proteins negatively regulate the signaling of the NF-κB and STAT/JNK pathways [132,133,134,135].	As for NK cells, it has been found that the combined inhibition of IL-1R8 and exposure to IL-15 enhances the functional activity of these cells through the activation of the Jak/STAT and PI3K/AKT signaling pathways. This activation leads to an increase in the synthesis and secretion of IFNγ, GM-CSF, CCLS, CXCL8, as well as granzyme B. As a result, the antitumor activity of NK cells is significantly increased [146,147,148,149].	-
KIR (inhibitory receptors)	NK cells	The cytoplasmic tail contains several intracellular inhibitory ITIM domains. The inhibitory signal from these domains is carried out through a mechanism that involves phosphorylation of the ITIM domain, as well as the involvement of the phosphatases SHP1 and SHP2. These phosphatases disrupt the functional activity of molecular pathways such as PI3K/AKT and MAPK, as well as NF-kB [51,105,150].	The effects of inhibiting the PI3K/AKT, MAPK, and NF-κB pathways are discussed above.	00552396 (2007); 01256073 (2007); 00552396 (2007); 00999830 (2009); 01248455 (2010); 01222286 (2010); 01217203 (2010);
Siglec-7	Granulocytes, monocytes, NK cells, dendritic cells, T lymphocytes	It contains an intracellular ITIM domain on the cytoplasmic tail, which transmits an inhibitory signal through a mechanism involving phosphorylation of the ITIM domain and the participation of phosphatases SHP1 and SHP2. This disrupts the functional activity of molecular pathways such as PI3K/AKT and MAPK, and NF-κB [66,151].	The effects of inhibiting the PI3K/AKT, MAPK, and NF-κB pathways are discussed above.	-

* Human Protein Atlas (HPA) RNA sequencing data [27]. ** HPA data, based on the results of Gianni Monaco.

**Table 3 cells-14-01858-t003:** Clinical trials for the treatment of cancer using genetically modified NK cells.

Group Name	ID	Disease	Genetic Modifications	Target	Modification Properties
“Classic” CAR-NK cells, targeted at specific tumor antigen	NCT06454890 (2024–to the present)	Non-small cell lung cancer	antiTROP2-CAR-NK	TROP2	TROP2 is a transmembrane protein that is expressed in various types of epithelial tumors. It activates the MAPK and PI3K/AKT signaling pathways, which are involved in cell growth, survival, and metastasis. antiTROP2-CAR-NK cells eliminate TROP2-expressing cells [160].
	NCT06201247 (2023–to the present)	Acute myeloid leukemia	antiCD123-CAR-NK	CD123	CD123 is expressed at moderate levels on CD34+ hematopoietic precursor cells, and its expression significantly increases with the development of hematolymphoid neoplasms, including acute myeloid leukemia. antiCD123-CAR-NK cells eliminate CD123-positive cells [161,162]
	NCT05645601 (2022–2024)	Refractory B-cell hematological malignancies	antiCD19-CAR-NK	CD19	CD19 is ubiquitously expressed by B cells at all stages of their differentiation. Its expression significantly increases with the development of B-cell malignancies. Therefore, antiCD19-CAR-NK cells specifically target and eliminate CD19-positive cells [163].
	NCT06045091 (2023–to the present)	Relapsed/refractory multiple myeloma and plasma cell leukemia	antiBCMA-CAR-NK	BCMA	BCMA is involved in the proliferation and differentiation of B cells. Its expression can increase with the development of malignant blood diseases, especially with the development of multiple myeloma. Due to this, BCMA has been considered one of the promising markers for the treatment of this disease. antiBCMA-CAR-NK cells are designed to specifically eliminate BCMA-positive cells [164].
	NCT06696846 (2024–to the present)	Acute myeloid leukemia	antiCD70-CAR-NK	CD70	CD70 is a transmembrane protein belonging to the TNF family that is expressed on acute myeloid leukemia blast cells. Unlike CD123, it is not expressed in normal tissues, making it a promising target for immunotherapy. antiCD70-CD70 CAR-NK cells can specifically eliminate CD70-positive cells [165].
	NCT05507593 (2022–2023)	Small-cell lung cancer	antiDLL3-CAR-NK	DLL3	DLL3 is overexpressed during the development of small-cell lung cancer at various stages and promotes cell proliferation, modulates the microenvironment, and contributes to resistance to the immune response. It inhibits Notch signaling pathway and is activated by ASCL1 transcription factor. In normal cells, its expression is low. antiDLL3-CAR-NK cells are designed to specifically eliminate DLL3-positive cells [166,167].
	NCT02839954 (2016–2018)	Solid tumors, such as MUC1+ (Mucin short variant S1, or Mucin 1) malignant glioma of the brain, colorectal carcinoma, gastric carcinoma, hepatocellular carcinoma, non-small cell lung cancer, pancreatic carcinoma, and breast carcinoma.	antiMUC1-CAR-NK	MUC1	MUC1 (mucin) is a transmembrane protein that is highly glycosylated and normally forms a protective layer on the surface of epithelial cells. However, in certain types of tumors, its expression significantly increases and it changes its localization within cells. Additionally, its glycosylation becomes incomplete. MUC1 plays a role in tumor metastasis, apoptosis regulation, and formation of resistance to the immune response. antiMUC1-CAR-NK cells are designed to specifically eliminate MUC1-positive cells [168].
	NCT05194709 (2021–2022)	Solid tumors	anti5T4-CAR-NK	5T4	5T4 (trophoblast glycoprotein) is a transmembrane protein that is normally expressed in placental cells and plays an important role in fetal survival. However, it is almost never expressed in normal adult tissues. Its overexpression has been linked to the formation of various types of solid tumors. Evidence is also emerging that 5T4 may be overexpressed in the development of certain hematological malignancies. anti5T4-CAR-NK cells are designed to specifically target and eliminate 5T4-positive cells [169,170,171].
CAR-NK cells of “universal action”	NCT05247957 (2021–2022)	Acute myeloid leukemia	CAR-NK-NKG2D	NKG2DL (MICA, MICB, ULBP1-6)	NKG2D is an activating receptor of NK cells. It is expressed in more than 80% of tumor types, but it is practically not expressed in normal cells. Therefore, CAR-NK-NKG2D can be used to treat a wide range of tumors [29].
	NCT03415100 (2018–2019)	Metastatic solid tumors			
	NCT05213195 (2021–to the present)	Refractory metastatic colorectal cancer			
	NCT06478459 (2024–to the present)	Pancreatic cancer			
	NCT05776355 (2023–2024)	Ovarian cancer			
“biCAR”-NK cells that target two antigens at once, or CAR-NK cells with multiple genetic modifications	NCT06652243 (2024–to the present)	Hepatocellular carcinoma	antiGPC3-CAR-NK + secreted IL15	GPC3	GPC3 (glypican 3) is a protein that is associated with the cell membrane and belongs to the GPC family. There are 6 types of glypicans in this family, including GPC1-GPC6. GPC3 is expressed in the ovaries and embryo cells, but its expression has not been observed in other tissues. During the development of liver cancer, however, the expression of GPC3 increases significantly, making it a promising target for immunotherapy. antiGPC3-CAR-NK cells are designed to specifically eliminate GPC3-positive cells [172,173]. IL-15 plays a critical role in the development of NK cells, from early precursors to mature NK cells. In mature NK cells, the transition from the CD56^bright^ to CD56^dim^ phenotype is facilitated by the interaction with IL-2 and IL-15. Therefore, IL-15 secreted by CAR-NK cells can further activate the antitumor activity of NK cells in the patient’s body, as well as support the CAR-NK cells that are injected into the patient [71].
	NCT06342986 (2024–to the present)	Ovarian, fallopian tube, and primary peritoneal cancer	antiMICA/B-CAR-NK + *CD38* knockout, high-affinity, non-cleavable CD16 (hnCD16), IL-15 with IL-15R expression (IL15/IL15R)	MICA, MICB	MICA and MICB are ligands for the NK cell-activating receptor NKG2D. They are expressed in more than 80% of various types of malignant neoplasms. It has also been noted that MICA and MICB are expressed by ovarian cancer tumor cells, but they are not expressed in normal tissues except for epithelial cells. antiMICA/B-CAR-NK cells specifically eliminate MICA- and MICB-positive cells [174]. The CD38 receptor is expressed on immune system cells, particularly NK cells. To protect NK cells from therapeutic antibodies targeting CD38, a knockout can be introduced into the *CD38* gene, avoiding a “fratricidal” reaction between NK cells. A genetically modified version of the CD16 receptor, hnCD16, has increased affinity for the Fc fragment of antibodies and resistance to ADAM17 metalloproteinase. This is achieved by modifying the molecular structure of the protein by replacing phenylalanine with valine at position 158 and serine with proline at position 197 [175,176].
	NCT05182073 (2021–to the present)	Multiple myeloma	antiBCMA-CAR-NK + *CD38* knockout, hnCD16 expression, IL15/IL15R expression	BCMA	BCMA, CD38 and CD16 receptors—see above in the table.
	NCT05987696 (2023–to the present)	Acute myeloid leukemia	antiCD33-CAR-NK + CLL1 secretion	CD33	CD33 is a transmembrane protein that is expressed on the surface of myeloid lineage cells. It plays a role in cell adhesion and the transmission of intercellular signals. It can be found on blast cells in acute myeloid leukemia. antiCD33-CAR-NKT purposefully eliminate CD33-positive cells [164,177,178]. CLL1 is a chemokine that interacts with the CCR8 receptor to attract immune cells to the inflammation site and activate them. Thus, the expression of CLL1 chemokine by antiCD33-CAR-NK cells can further attract the patient’s own immune cells to the tumor site. This can enhance the cytotoxic effect against cancer cells [179,180].
	NCT06594211 (2024–to the present)	Multiple myeloma	antiBCMA/GPRC5D-CAR-NK (biCAR)	BCMA and GPRC5D	GPRC5D is a transmembrane protein that is typically expressed in plasma and epithelial cells. It is highly expressed in multiple myeloma cells, and its function is not yet fully understood [181]. However, this protein has shown promise as a potential target for treatment of multiple myeloma. antiBCMA/GPRC5D-CAR-NK cells can specifically eliminate BCMA- and GPRC5D-positive cells [164]. Bispecific CARs may increase the cytotoxicity of these CAR-NK cells against tumor cells and improve their selectivity for target cells.
NK cells with other genetic modifications	NCT04991870 (2023–to the present)	Glioblastoma	NK cells with *TGF-βR2* and *NR3C1* knockout	Glioblastoma cells	TGF-β is a cytokine secreted by cancer cells that has immunosuppressive properties. It helps reduce the body’s antitumor response while simultaneously forming the tumor microenvironment and promoting metastasis. The inhibition of TGF-β signaling through the knockout of its receptor is a promising immunotherapy strategy [182]. *NR3C1* encodes the glucocorticoid receptor, and it has been found that the binding of glucocorticoids to this receptor leads to a significant decrease in the functional activity of NK cells. The knockout of this gene may lead to increased antitumor cytotoxicity in CAR-NK cells [183,184].

**Table 4 cells-14-01858-t004:** Targeted tumor antigens and related clinical trials.

Disease	Target	Clinical Trials
B-Cell Non-Hodgkin Lymphoma	CD19	NCT06707259, NCT06334991, NCT05842707, NCT05739227, NCT06464861, NCT05020678 NCT03824964
	CD22	NCT03824964, NCT03692767
	CD70	NCT05842707, NCT05092451
B-lymphoblastic leukemia (B-ALL)	CD19	NCT06631040, NCT05739227, NCT05563545, NCT05020678
Blastic plasmacytoid dendritic cell neoplasm	CD123	NCT06690827, NCT06006403
Hepatocellular carcinoma	GPC3 (glypican 3)	NCT06652243,
	CD70	NCT05703854
	MUC1 (mucin 1)	NCT02839954
Glioma	MUC1 (mucin 1)	NCT02839954
Breast carcinoma	MUC1 (mucin 1)	NCT02839954
Endometrial cancer	Claudin 6	NCT05410717
	GPC3	NCT05410717
	Mesothelin	NCT05410717
Castration-resistant prostate cancer	PSMA	NCT03692663
Colorectal cancer	TROP2	NCT06358430
	MUC1 (mucin 1)	NCT02839954
Mantle cell lymphoma (MCL)	CD5	NCT05110742
	CD19	NCT06464861, NCT05020678
Hodgkin’s lymphoma	CD70	NCT05092451
Central Nervous System Lymphoma	CD19	NCT06827782
Mesonephric-like adenocarcinoma	TROP2	NCT05922930
Mesothelioma	CD70	NCT05703854
Myelodysplastic syndrome	CD33	NCT06325748
	CD70	NCT05092451
	FLT3	NCT06325748
Small-cell lung cancer	DLL3	NCT05507593
Multiple myeloma	CD70	NCT05092451
	BCMA	NCT06594211, NCT06242249, NCT06045091, NCT05652530, NCT05182073
	GPRC5D	NCT06594211
Non-small cell lung cancer	TROP2	NCT06454890
	MUC1 (mucin 1)	NCT02839954
Osteosarcoma	CD70	NCT05703854
Acute myeloid leukemia (AML)	CD7	NCT02742727
	CD33	NCT06325748, NCT05987696, NCT05215015, NCT05008575
	CD70	NCT06696846, NCT05092451
	CD123	NCT06690827, NCT06201247, NCT06006403, NCT05574608
	FLT3	NCT06325748
Primary mediastinal B-cell lymphoma (PMBCL)	CD19	NCT06464861
Peritoneal carcinomatosis	MICA, MICB	NCT06342986
Plasma cell leukemia	CD70	NCT05092451
	BCMA	NCT06045091
Gastric cancer	Claudin18.2	NCT06464965
	MUC1 (mucin 1)	NCT02839954
Pancreatic cancer	Claudin18.2	NCT06464965
	TROP2	NCT05922930
	ROBO1	NCT03941457
	MUC1 (mucin 1)	NCT02839954
Fallopian tube cancer	MICA, MICB	NCT06342986
Ovarian carcinoma	MICA, MICB	NCT06342986
	TROP2	NCT05922930
	Claudin 6	NCT05410717
	GPC3	NCT05410717
	Mesothelin	NCT05410717, NCT03692637
Adult T-cell leukemia/lymphoma (ATLL)	CD5	NCT06909474, NCT05110742
	CD7	NCT06849401, NCT02742727
	CD19	NCT05563545
	CD70	NCT06696846, NCT05092451

## Data Availability

The datasets used and/or analyzed during the current study are available from the corresponding author upon reasonable request.

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
