# Peer review of "Strategies for the Development of NK Cell-Based Therapies for Cancer Treatment"

_cells, 2025, doi:10.3390/cells14231858_

Round 1
Reviewer 1 Report
Comments and Suggestions for Authors
In this review, the authors discussed existing approaches to modifying CAR-NK cells and assessed the results of clinical trials involving CAR-NK therapies. Conventional approaches to NK cell modification can be divided into three main groups: strategies to enhance antitumor cytotoxicity, strategies to improve the survival of CAR-NK cells and prolong their persistence in the body, and strategies to increase the safety of CAR-NK cells. Although this review has some limitations, it is helpful in future directions for the development of innovative CAR-NK therapies. Overall, this review paper is very interesting, but there are still several issues to be addressed before its acceptance.
Major comments:
- Single therapies including CAR-NK treatment struggle to address the complex network of tumor immune suppression, making combination therapy a key pathway to improve efficacy. The authors should also provide some discussions on which type of drugs is preferred to be in combination with CAR-NK in cancer treatment.
- Except the sections mentioned by the authors, optimization of metabolic function in CAR-NK also plays a critical role in enhancing its efficacy. The authors should add some discussions in the review.
- In future development directions, dual/multiple-specific CARs targeting NK therapies will be developed to reduce antigen escape, which should be discussed in the manuscript.
Reviewer 2 Report
Comments and Suggestions for Authors
This is excellent work on this topical and important suject. The subject is well thoght through and the manuscript well versed.
I think a can spot only one mistake. Om page 8, line 315, I think "valine" should be replaced by "serine".
Reviewer 3 Report
Comments and Suggestions for Authors
This is an excellent review article describing current innovations for the development of cancer-specific and universal NK cell-based immunotherapies. Only a few minor comments:
- There are certain themes repeated in the manuscript. Removing them may reduce the manuscript length.
- Lines 135-137: Not clear how tumor cells decrease anti-apoptotic proteins in immune cells; provide reference.
- Line 406: MSLN?
- Line 437: what is dratumumab’s target?
- CAR-NK cells of "universal action": NKG2DL is expressed on tumor cells? Please verify and correct if needed.
- NK cells with other genetic modifications: NK cell with genes encoding or knocked out?
- Consider eliminating Table 4, or modifying it in a better way.
Reviewer 4 Report
Comments and Suggestions for Authors
This review centers around genetically modified NK cells as immune engagers for cancer therapy. One of the major advantages of the article is that it does a very good job of describing most of the important genetic modifications designed to address improve CAR-NK cytotoxicity, safety and longevity. Also, the tables do address important issues of clinical trials and varied targets. One of the disadvantages is that in 2025 several similar reviews have already been published. For example,
Beyond CAR-T: Engineered NK cell therapies (CAR-NK, NKCEs) in next-generation cancer immunotherapy. Zhang F, Soleimani Samarkhazan H, Pooraskari Z, Bayani A.Crit Rev Oncol Hematol. 2025 Oct;214:104912.
and
Natural killer cell-based immunotherapy for cancer. Ma S, Yu J, Caligiuri MA. J Immunol. 2025 Jul 1;214(7):1444-1456.
Points that need to be addressed in the paper are as follows:
-lines 73-75. The authors state "There are three main types of NK cells, based 73 on the expression of different surface markers: CD56bright, CD16dim, CD57neg; 74 CD56dim, CD16bright, CD57neg; CD56dim, CD16bright, CD57pos." But what type of these types of NK cells are used to make CAR-NK?
-line 91. Isn't the CD56dimCD16dimCD57pos population of memory NK cells another type of NK cell?
-line 176. Eliminate the redundancy. "MICA (MHC class I polypeptide–176 related sequence"
-line 201. When you talk about the hinge domain, it would be helpful to have a diagram of the interior of the CAR NK.
line 291. No need for a new paragraph
Figure 1. You can save space by making the Figure smaller and the labeling larger.
line 375. Stating that "CAR-NK cell therapy has revolutionized the treatment of cancer." is overstating." This technology is not yet clinically proven.
-line 435,436. You need to clarify your point with CD38. its too vague.
-lines 465-474. Needs to be more clearly written.
-Conclusion paragraph is weak.
Reviewer 5 Report
Comments and Suggestions for Authors
In the review paper by Budagova and coauthors an up-to-date and very comprehensive overview is provided regarding various established but also novel and more experimental strategies to combat cancer by using NK cells modified with chimeric antigen receptors. The description of genetic modifications in CAR cell therapy including variations of scFv antibodies in the targeting domains of CARs, different signaling domains in CARs, NK-cell cytotoxicity and survival enhancing features such as non-cleavable CD16 variants, membrane-bound IL15/IL15RA and the knock-out of genes contributing to NK cell inhibition, is interesting to read. The review strongly benefits from the comprehensive list of clinical trials involving CAR-modified NK cells.
Minor points:
The parts 2.1 "NK cell biology" and 2.2 "Cancer progression" are somewhat superficial and poorly documented by citations. These parts seems to be beyond the scope of the review and could be omitted or significantly shortened.
Lines 173-177: Be more specific about the expression patterns of NKG2D ligands.
Line 184: Sentence is repeated.
Line 193: Be more specific about the expression of BCMA.
Line 205-207: The stalk domain of CD8A/B can homo/heterodimerize rather than CD28 the CD28 transmembrane-domain.
Lines 430-437: This paragraph is confusing. In my understanding the use of daratumumab impairs NK cell treatment of multiple myeloma treatment with NK cells as CD38 is not only expressed by MM tumor cells but also by NK cells themselves, thereby inducing fratricide.
Lines 305-306: Be more specific about the locations of ADAM17 cleavage sites. This enumeration is partly incorrect.
Line 358 and whole paragraph: "through transpresentation". It should be made clear that the expression of mbIL-15/IL-15RA is an artificial approach that is not available in normal NK cells.
Lines 638-641. The use of the NKG2D ectodomain in CAR-like constructs should be explained in more detail and discussed in the context of NKG2D ligand shedding by tumor cells.
Lines 597-617: Cite appropriate publications for the rescue epitope RQR8 and for tEGFR and not BMS product advertisement (ref. 147).
Tables 3 and 4 are partially redundant.
References 12, 19 and 22 are inappropriate.
Round 2
Reviewer 1 Report
Comments and Suggestions for Authors
Following the authors’ responses, the revised manuscript has been sufficiently improved to warrant publication in Cells.
Reviewer 4 Report
Comments and Suggestions for Authors
Revisons are adequate.